# Conceptual Model for the Vulnerability Assessment of Springs in the Indian Himalayas

**Denzil Daniel** [1] , **Aavudai Anandhi** [2] **and Sumit Sen** [1,3,*]

1 Centre of Excellence in Disaster Mitigation and Management, Indian Institute of Technology Roorkee, Roorkee 247667, India; denzil.dm@sric.iitr.ac.in

2 Biological Systems Engineering Program, College of Agriculture and Food Sciences, Florida A&M University, Tallahassee, FL 32307, USA; anandhi.swamy@famu.edu

3 Department of Hydrology, Indian Institute of Technology Roorkee, Roorkee 247667, India

* Correspondence: sumit.sen@hy.iitr.ac.in; Tel.: +91-1332-284754

**Abstract:** The Indian Himalayan Region is home to nearly 50 million people, more than 50% of whom are dependent on springs for their sustenance. Sustainable management of the nearly 3 million springs in the region requires a framework to identify the springs most vulnerable to change agents which can be biophysical or socio-economic, internal or external. In this study, we conceptualize vulnerability in the Indian Himalayan springs. By way of a systematic review of the published literature and synthesis of research findings, a scheme of identifying and quantifying these change agents (stressors) is presented. The stressors are then causally linked to the characteristics of the springs using indicators, and the resulting impact and responses are discussed. These components, viz., stressors, state, impact, and response, and the linkages are used in the conceptual framework to assess the vulnerability of springs. A case study adopting the proposed conceptual model is discussed for Mathamali spring in the Western Himalayas. The conceptual model encourages quantification of stressors and promotes a convergence to an evidence-based decision support system for the management of springs and the dependent ecosystems from the threat due to human development and climate change.

**Keywords:** vulnerability assessment; Indian Himalayas; springs; springshed management; water security

## 1. Introduction

Springs in the mid-hills of the Hindu Kush Himalayas (HKH), crucially important for the survival of the 240 million hill and mountain people residing in the region [1], are drying [2–7]. The mid-hills are located at a lower elevation compared to the largest reserves of snow and ice outside the North and South Poles (the HKH sometimes referred to as the 'Third Pole'). Though the 10 rivers that originate in the glaciers of these 'water towers' benefit the nearly 2 billion people living in their hydrological downstream, this rich supply of freshwater reserves is out of reach for the mid-hill communities. These communities are dependent primarily on water stored underground and naturally drained through springs for their domestic and irrigation needs.

The paradoxical role of the humble springs, in contrast to the vast glaciers and rich rivers of the HKH, is not unusual considering the distribution of water on the surface of the 'Blue Planet'. Most of the water covering the Earth's surface, as seen from outer space, nearly 71% (hence the sobriquet 'Blue Planet'), is practically useless, with the actual reserves of freshwater accounting for only close to 2.5% of the total water of the world and the remaining reserves of water trapped in the world's oceans, saline lakes, and deep underground reservoirs [8]. Usable groundwater accounts for about 30% of these freshwater reserves, while flow in rivers, water in swamps, and freshwater lakes, together, make up a little over 0.03% [8]. Most of the entire groundwater reserves drain

into springs and streams at different time scales, ranging from less than a year to more than 100,000 years [9]. Going by these broad estimates, natural springs, which are freshwater resources at the interface of surface water and groundwater, where underground water emerges on to the Earth's surface, forming streams, ponds, and swamps [10], account for even less than 0.03% of the freshwater resources of the world at any given time.

Though seemingly small in terms of absolute volume of reserves of freshwater, springs have been a vital source of water for drinking and sanitation since ancient times (for example, Roman, Mesopotamian, Egyptian, and Chinese civilizations developed systems to tap and transport spring water) [11]. Even by recent estimates, water from karst springs, which are some of the largest springs of the world, provides drinking water to 678 million people, or 9.28% of the world's population [12]. The World Health Organization estimated in 2017 that 435 million people still draw water directly from unprotected wells and springs [13]. Though the share of springs in the global freshwater reserves is small, the dependence for water primarily on springs poses a water security threat in many parts of the world, especially the HKH. In this paper, we draw attention to the Indian Himalayas, a smaller geographical extent within the HKH, but the lessons learnt apply to the larger HKH region as well.

In the Indian Himalayan Region (IHR), springs are the main source of water for rural communities [14–16]. Both rural and urban settlements, with a combined population of nearly 50 million, depend on water sourced from springs for use in drinking, domestic, livestock, and agricultural water needs [14,17–19]. There are 60,000 villages, 500 growing townships, and 8–10 cities in the IHR, and 23.9% of all surface water schemes in IHR states are sourced from springs, with more than 20% of villages in IHR states reporting having a spring [14]. Even by conservative estimates, more than 50% of the population in the IHR, thus, are dependent on spring water for their sustenance [14]. Additionally, out of the 12.5% of cultivated area in the IHR, only 11% is irrigated, of which 65% is sourced from springs [14]. Though these numbers are a gross underestimation and are based on secondary data, these numbers represent the dependence of a large proportion of the population in the IHR on water sourced from springs [14]. IHR springs are also the source of the revered Himalayan rivers, such as headwater tributaries of the River Ganga and River Yamuna River [14]. Though a lack of studies that estimate the exact contribution of the Himalayan springs to the rivers originating in the Himalayas exists, there is some evidence to suggest that flow in these rivers during lean seasons is maintained, to a large extent, by contributions from springs and groundwater flow [2,6,14,20]. In several regions, springs are considered sacred, holding cultural and religious significance to communities dependent on them [3,21]. However, in recent decades, nearly 50% of perennial springs and 60% of low-discharge springs have shown a decline in discharge, if not being fully dried up [14].

In recent decades, a reduction in spring discharge and declining water quality have been observed in separate studies across the IHR (for example, Central Himalaya [2], Western Himalaya [4]). Isolated studies in different parts of the IHR report population growth [7,15], development activities (road construction) [4,15,22] and rapid urbanization [7], climate change and variability [6,21,23], and land use land cover changes [4,7] as reasons for the decline in spring discharge, and unplanned urban development for the poor water quality [24,25]. The author of [17] presents a synthesis of the effect of anthropogenic activities and climate change on springs and suggests measures to mitigate them. The impact of the drying of springs on the water security of the region is further aggravated by demand-side pressures on water resources. The Himalayas have seen an increase in the number of urban centers, an increase in the urban population due to migration of people from rural and semi-urban regions to these urban centers, and an increase in the seasonal floating population consisting of recreational and religious tourists [19]. The growth in the urban population has led to exploitation of water resources in the region, with urban centers dependent on water from springs and rivers through piped networks [26]. The

increasing demand and decreasing supply of freshwater from springs are a water security threat in the Himalayan region.

Led by an urgent need to unify efforts across the IHR at the national level, the National Institution for Transforming India [14] (NITI Aayog), a policy 'Think Tank' of the Government of India, constituted a working group in 2017 to assess the magnitude of the problem and synthesize policies, initiatives, and best practices from across the IHR [14]. With an estimated 3 million springs in the IHR alone [14], there exists a need for a framework to identify springs and regions in the IHR that are most vulnerable and thus need urgent attention for rejuvenation, restoration, and better management. This study aims at providing a conceptual framework to assess the vulnerability of the Himalayan springs. In the following sections, after defining the study region, we begin conceptualizing vulnerability (concept) to stressors in springs in the Indian Himalayas. This is followed by a systematic review of previously published research on each component of vulnerability assessment (operational procedures). The conceptual model is discussed through a case study of Mathamali spring in the Western Himalayas.

Vulnerability in the context of management of springs and other natural resources is not entirely novel in India. The National Rainfed Area Authority (NRAA) of the Ministry of Agriculture and Farmer Welfare, Government of India, developed a composite index for the prioritization of districts for development planning [27]. The 670 districts of the country, representing more than 90% of the population and geographical area, are ranked based on a composite index computed as a weighted aggregate of two rescaled indices (natural resource index and livelihood index), which are themselves weighted aggregates of 12 and 18 parameters, respectively. Similarly, a village-level assessment of climate-related vulnerability has been undertaken in Sikkim [21] using an exposure, sensitivity, and adaptive capacity framework. The drying up of springs is reported as an impact of climate change on the water resources of villages [15,21], and hence intensive studies on reviving dying springs were subsequently taken up in the drought-prone west and south districts identified based on vulnerability analysis at the village level. Both these studies prioritize development activities within administrative boundaries using an index-based approach. In the present study, we look at springs and clusters of springs as the unit of study. Inventorying of springs has been taken up in recent years in many states by governments (e.g., the Sikkim spring atlas developed by the Rural Development Department, Government of Sikkim, https://sikkim-springs.gov.in/spring_atlas, accessed on 9 June 2021) and non-governmental agencies (e.g., the spring atlas of Uttarakhand developed by CHIRAG, an NGO in Uttarakhand, http://chirag.org/wp-content/uploads/2019/07/SPRING-ATLAS-CHIRAG.pdf, accessed on 9 June 2021). Therefore, a quantitative measure of the vulnerability of springs will augment the inventory exercise to enable targeting of springs and clusters of springs requiring immediate attention for rejuvenation.

## 2. Study Region

The IHR comprises ten (10) states and four (4) hill districts of India stretching in an East–West orientation for 2500 km along the northern border of India. This region covers three (3) biogeographic zones—the Himalayas, the Trans Himalayas, and the northeast hills. A biogeographic zone is a large distinctive land unit of similar ecology, biome representation, community, and species [28] (the biogeographic classification of India was first published as a draft in 1988 by the Wildlife Institute of India (WII) and further refined and published in the review of the protected areas network for India). In developing a framework for vulnerability assessment, this paper focuses on the Himalayas biogeographic zone spread across five Indian states (Jammu and Kashmir, Himachal Pradesh, Uttarakhand, Sikkim, Arunachal Pradesh) and two hill districts of West Bengal (namely, Darjeeling and Kalimpong). The Himalayas further consist of four biotic provinces (North-West, West, Central, and East Himalayas). The North-West and West Himalayas are in the states of Jammu and Kashmir, Himachal Pradesh, and Uttarakhand. The central province comprises Sikkim and the hill districts of West Bengal, while Arunachal Pradesh is in East Himalaya.

In this paper, regions west of Nepal are considered the western region and the states east of Sikkim as the eastern region. Only Sikkim and the hill districts of West Bengal are in the central region of the Himalayas. The Kumaon region of Uttarakhand is sometimes considered as the Central Himalayas (e.g., [4]), but in this paper, the whole of Uttarakhand will be considered part of the western region of the Indian Himalayas.

## 3. Conceptualizing Vulnerability in Springs in the Indian Himalayas

In this study, we developed a conceptual model to represent vulnerabilities in the Indian Himalayan springs by adapting the triple complexity for adaptive management systems proposed in [29] and the WR-VISTA framework proposed in [30]. For this purpose, vulnerabilities in the Himalayan springs to stressors are considered a connected system consisting of the (1) complexity in water management and planning systems, (2) complexity of spring systems, and (3) complexity of stressor systems (Figure 1a,b). To represent this triple complexity, we need to translate stressors to impacts in the Himalayan springs. Top-down, bottom-up, and hybrid approaches can be useful for this (Figure 1a). Then, we represent the degree of stressors and springs using measured and/or surveyed information (variables and/or indicators; e.g., temperature, rainfall, water yield, farmer choice), and water management is represented using concepts (vulnerability). Finally, we translate the variables and/or indicators to concepts (Figure 1b) using four pathways (Figure 1c). Traditional studies use Path 1 (variable-model-vulnerability) and/or Path 2 (variable-vulnerability).

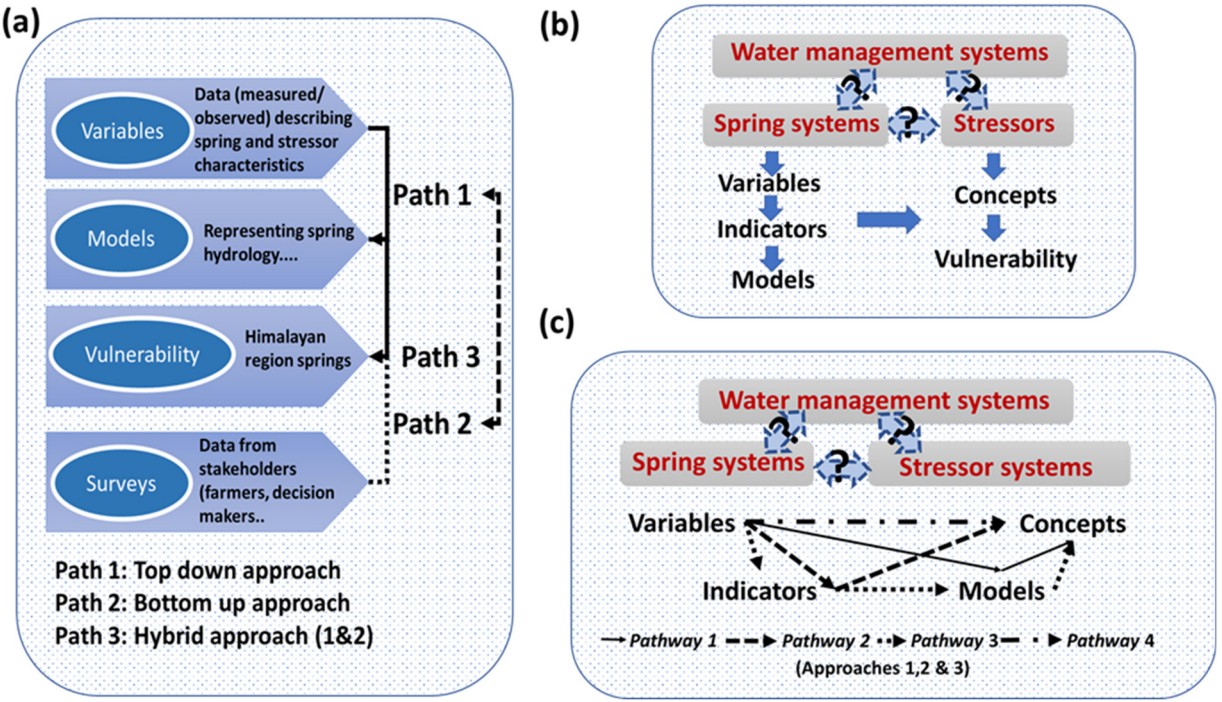

**Figure 1.** Vulnerability of Himalayan springs to variable and changing stressors. (**a**) Three approaches (based on starting points) that can be used in vulnerability assessments. (**b**) Systems approach to represent the triple complexity. (**c**) Approaches and pathways to representing the complexity in IHR. In (**b**) and (**c**), the inter-connectedness between the three systems is represented using dashed-bordered arrows. The solid-bordered arrows in (**b**) represent the general information flow while translating variables to the vulnerability concept. The first column of solid-bordered arrows (left-hand side) in (**b**) represents the water management systems and stressor systems, while the second column of solid-bordered arrows (right-hand side) in (**b**) represents the vulnerability of the water management system. The information flow through solid-bordered arrows in (**b**) is divided into four pathways using 4 arrow types in (**c**).

In our attempt to develop quantitative vulnerabilities of the Himalayan springs to stressors using indicators and the system's approach, we introduce two other potential pathways (Pathways 2 and 3, Figure 1c). For this, first, we developed a conceptual model

incorporating the systems approach which is based on systems theory, for vulnerability assessment. The systems theory [31] states that complex, nonlinear systems function differently in vivo than the separate scrutiny of their parts might indicate. The advantage of this study is derived from the systems approach's ability to extract information about the functioning of complex systems, which cannot be garnered from a sequence of isolated subsystem-scale studies [29].

The triple complexity we propose to be incorporated into the conceptual modeling framework process requires a holistic perspective rather than a reductionist perspective. Our problem has the common challenges of complex systems and data-scarce regions, including (1) multi-scale processes and multi-scale interactions; (2) emergent self-organization properties that reveal new behaviors; (3) multiple stressors and their interconnections; (4) limited understanding of processes; (5) theoretical concepts that cannot be measured directly; and (6) limited datasets to represent the processes and interactions. Quantifying issues enmeshed in multi-scaled connectivity is a fundamental characteristic of complex systems, and the process of the quantifying function within this complexity is the fundamental objective of the systems approach [32].

The developed conceptual framework identifies the stressors (internal/external), characterizes the processes in the springs in the Indian Himalayas, and synthesizes the impacts and responses from the literature and expert knowledge about the study region.

The top-down, bottom-up, and hybrid approaches are useful analytical starting points. Top-down approaches most frequently use measured/observed data (variables), which are then input to models that mathematically represent the system and take action, while in the bottom-up approach, information is collected using surveys, questionnaires, etc., from key stakeholders (e.g., producers, managers, planners) [29]. They can provide an appropriate starting point from an implementation perspective. Hybrid approaches also exist. These approaches utilize the stakeholder information from bottom-up models with the top-down models.

### 3.1. Typologies of Indian Himalayan Springs

A review of published research on springs in the Indian Himalayas reveals a diversity of springs in the region, and, consequently, these have been classified in numerous ways based on geology, hydrology, water chemistry, water temperature, human use, and the landscape of the recharge area (Table 1).

**Table 1.** Synthesis of typologies of Himalayan springs.

| Classification Type | Categories (Examples) | Region | Sources |
|---|---|---|---|
| *Geology* | | | |
| Bedrock | Triassic limestone, alluvium (flood plains), Karewas (lacustrine deposits), Panjal traps (volcanic rocks) | Western (Kashmir) | [23] |
| Geological structure (genesis and nature of water-bearing formation) | Fracture/joint-related springs, fault lineament-related springs, colluvial springs, springs originating in fluvial deposits, bedding plane-related springs, sill or dyke-related springs, karst springs, underwater springs | Western (Kumaon division, Uttarakhand) | [4] |
| Geo-structural controls | Thrust (lineament)-controlled, fault-controlled, bedding plane-controlled, fracture-controlled, joint-controlled, shear zone-controlled, fluvial deposit-related | Western (Nainital district, Kumaon division, Uttarakhand) | [6] |
| Based on the underlying geology | Depression, contact, fracture, karst, and fault springs | Western (Uttarakhand), Central (Sikkim), and Eastern (Arunachal Pradesh) | [2,15,33,34] |
| *Hydrology* | | | |
| Variation or persistence of flow | Springs with continuous flow, springs with interrupted flow, dead springs or springs that dried up very frequently | Western (Tehri-Garhwal district, Garhwal division, Uttarakhand) | [22] |
| Seasonality | Perennial springs (flow throughout the year), non-perennial springs (flow during some months), dry springs | Western (Nainital district, Kumaon division, Uttarakhand) | [6] |
| Based on spring hydrograph | Low discharge and highly seasonalWidely ranging discharge and seasonalHigh and perennial dischargeModerate and fairly constant and perennial discharge | Central (Sikkim) | [2] |

**Table 1.** *Cont.*

| Classification Type | Categories (Examples) | Region | Sources |
|---|---|---|---|
| *Water Chemistry* | | | |
| Water type (hydrochemical facies) | Ca–Mg–HCO3 water with low to moderate EC<br>Ca–Mg–HCO3–SO4 water with moderate EC | Western (Himachal Pradesh) | [35] |
| *Water Temperature* | | | |
| Temperature (with indirect reference to mean annual air temperature) | Cold springs (8–14 °C)<br>Warm springs (14–19°C) (MAAT is 12 °C) | Western (Kashmir) | [36] |
| Absolute temperature | Warm spring (25–37 °C)<br>Scalding spring (temperature >50 °C) | Eastern (Arunachal Pradesh) | [37] |
| *Human use* | | | |
| Based on suitability for irrigation | Classification based on salinity, sodium hazard (sodium absorption ratio (SAR)), soluble sodium percentage (SSP), residual sodium carbonate (RSC), magnesium index, permeability index, Wilcox diagram | Western (Almora, Kumaon division, Uttarakhand) | [18] |
| *Recharge area properties* | | | |
| Nature of the landscape comprising the catchment of the spring | Reserve forest (mixed), reserve pine forest, open pine forest, rainfed agriculture, irrigated land, population <5000, population 5000–10,000, population >10,000 | Western (Almora, Kumaon division, Uttarakhand) | [25] |

Classification of springs based on vulnerability to stressors will further spring the rejuvenation, restoration, and sustainable management agenda in the Indian Himalayas. In the next section, stressors are identified based on a synthesis of findings from peer-reviewed published studies.

### 3.2. Systematic Review of Stressors of Himalayan Springs

As per the 2011 Census of India, 46.96 million people live in the IHR (except Kalimpong in West Bengal which was demarcated as a separate district only in 2017) which is a decadal increase of 18.42% as compared to the 2001 census. The average annual exponential growth rate for IHR has decreased from 2.25 to 1.71 for the decades 1991–2001 and 2001–2011, respectively, but is still higher than the pan-India annual growth rate of 1.64. These figures are indicative of population pressure on the mountain ecosystems of the Himalayas. However, a closer look at studies on Himalayan springs indicates at least two broad categories of stressors: biophysical and socio-economic, each of which can be both external and internal (Figure 2). These drivers present a lens to map recent changes observed in the Himalayan springs that are affecting their sustainable use.

The stressor categories are defined based on the predominant effect of the stressor on the spring system. The biophysical drivers are the inter-related effects of land use and land cover change, climate change, and tectonic movements within the Earth's crust. These changes alter the recharge of springs and also affect storage and flow paths from the water source to the springs. The socio-economic drivers are the effects of a growing population and rapid human and economic development in the Himalayas. These drivers result in demand-side pressures while, at the same time, stifling supply side processes in the spring systems. Rapid development has also resulted in a deteriorating water quality in some springs due to the transport of anthropogenic pollutants to water sources that feed the springs. Table 2 presents a synthesis of drivers of change in Himalayan springs from published research articles. There exists a dearth of research on springs in the eastern region, with existing research (for example, [34,37]) only characterizing the springs in select locations in this region.

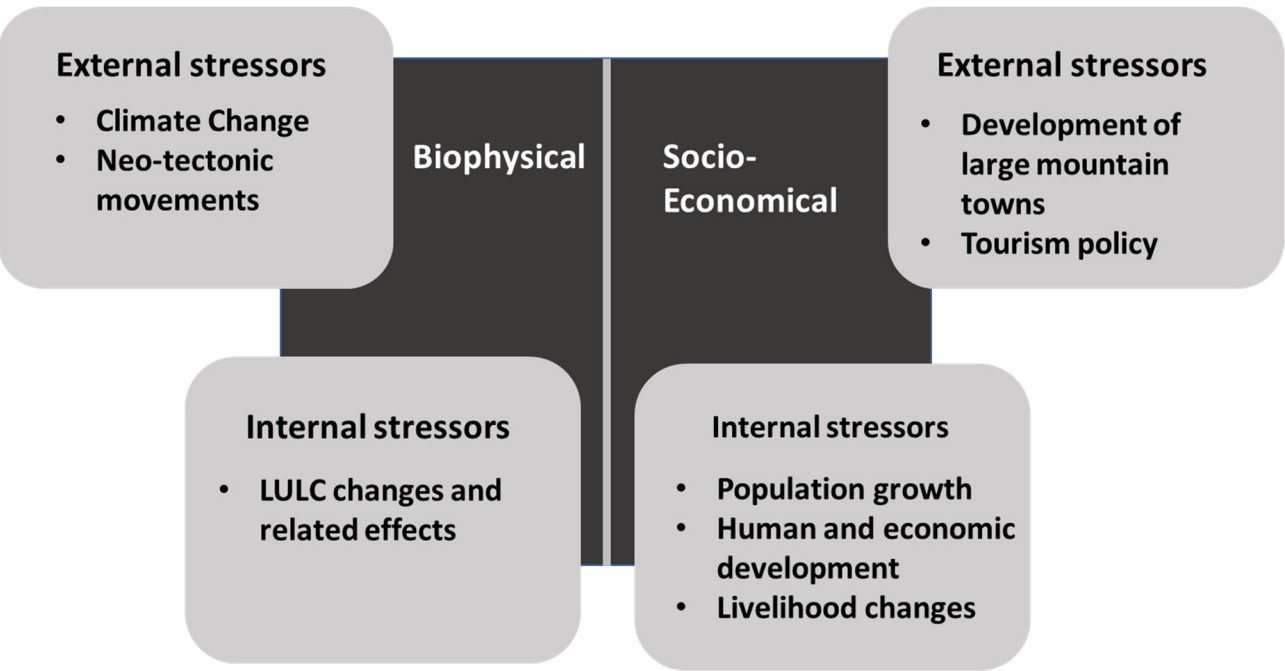

**Figure 2.** A conceptual representation of the challenges faced by the Himalayan springs: internal and external pressures and drivers impacting them.

**Table 2.** Synthesis of drivers of change in the Indian Himalayas.

|  | **Drivers** | **Sources** | **Region** |
|---|---|---|---|
| **A** | **Biophysical drivers** |  |  |
| A.1 | Internal |  |  |
| A.1.1 | Deforestation—forested land converted to barren land | [6] | Kumaon division, Uttarakhand, Western |
| A.1.2 | Replacement of multi-storied oaks with single-storied pine forests | [4,38] | Kumaon division, Uttarakhand, Western |
| A.1.3 | Change in the landscape in the recharge area—encroachment by invasive species (cacti and other xerophytes, Lantana camara, etc.) | [4] | Kumaon division, Uttarakhand, Western |
| A.1.4 | Slope failures and landslides | [6] | Kumaon division, Uttarakhand, Western |
| A.1.5 | Declining precipitation (particularly recycled precipitation recharging precipitation-fed springs) | [4] | Kumaon division, Uttarakhand, Western |
| A.1.6 | Forest fires | [15] | Sikkim, Central |
| A.1.7 | The decreasing trend of precipitation in the period of snow accumulation, i.e., November to February, resulting in rapid glacier retreat due to negative mass balance, affects glacier-fed springs | [23,39] | Kashmir, Western |
| A.1.8 | Geomorphological features of the recharge area (upslope area) of the spring | [16] | Garhwal division, Uttarakhand, Western |
| A.2 | External |  |  |
| A.2.1 | Precipitation decline due to climate change | [6] | Kumaon division, Uttarakhand, Western |
| A.2.2 | Reduction in the temporal spread of rainfall due to climate change, particularly decrease in winter rainfall | [15,21] | Sikkim, Central |
| A.2.3 | Increase in intensity of rainfall | [15,21] | Sikkim, Central |
| A.2.4 | Shifting of climate zones to higher altitudes resulting in reduction in oak forests and consequent replacement by pine and mixed forests | [6] | Kumaon division, Uttarakhand, Western |
| A.2.5 | Glacier retreat due to global warming and resulting negative mass balance | [23] | Kashmir, Western |
| A.2.6 | Neo-tectonic movements, active movements of tectonic plates resulting in changing thrusts, faults, and lineaments that affect groundwater flow | [6] | Kumaon division, Uttarakhand, Western |
| **B** | **Socio-economic drivers** |  |  |
| B.1 | Internal |  |  |
| B.1.1 | Livelihood practices—uncontrolled grazing, frequent plowing for multiple cropping, and higher constructional activity in the recharge area affect the water holding capacity of the soil | [25] | Kumaon division, Uttarakhand, Western |
| B.1.2 | Increasing fragmentation of landholdings. Therefore, springshed management requires greater coordination among more individuals who hold land in the recharge area. | [15] | Sikkim, Eastern |

**Table 2.** *Cont.*

|  | Drivers | Sources | Region |
|---|---|---|---|
| B.1.3 | Development in the vicinity of the spring | [22] | Garhwal division, Uttarakhand, Western |
| B.1.4 | Road widening, excavation for roads | [4,6,22] | Garhwal and Kumaon division, Uttarakhand, Western |
| B.1.5 | Expansion of canal network | [4] | Kumaon division, Uttarakhand, Western |
| B.1.6 | Upslope cutting for settlement and big buildings | [6,22] | Garhwal and Kumaon division, Uttarakhand, Western |
| B.1.7 | Encroachment into forest land for horticulture and agriculture | [4,6] | Kumaon division, Uttarakhand, Western |
| B.1.8 | Industrialization, urbanization | [4] | Kumaon division, Uttarakhand, Western |
| B.1.9 | Increasing population density in urban settlements, unplanned expansion, improper disposal of domestic sewage results in contamination of springs | [24,25,40] | Kumaon division, Uttarakhand, Western |
| B.1.10 | Out-migration and urban migration of men resulting in the feminization of springs management that is challenging local institutions | [41] | IHR |
| B.2 | External | | |
| B.2.1 | Development of densely populated settlements and large mountain towns in the same catchment as the springs under consideration, resulting in increasing demand for water | [5] | Kumaon division, Uttarakhand, Western |
| B.2.2 | Adoption of tourism models in the IHR that are not appropriate for mountain ecosystems. | [41] | IHR |

### 3.3. Characterizing the Stressors and the Springs in the Indian Himalayas

Stressors and springs have various characteristics such as properties, behaviors, internal processes, and position in the environment. For example, we can synthesize the typologies or classifications that use similarities of form and function to impose an order on springs. Basically, they are intellectual constructs in which objects with similar relevant attributes are grouped to represent the stressors and the springs. As evident from the schema of typologies of springs, a wide variety of information is collected related to springs which is then used in characterizing the springs. In an attempt to present a unified list, in this section, we follow a protocol for springshed management used successfully in the inventorying, rejuvenation, and management of springs in India. The authors of [33] presented, as an outcome of a consultative process with key stakeholders, a six-step protocol for the revival of springs in the Hindu Kush Himalayas drawing from previous experience in the implementation of spring and springshed management in Sikkim State of India under the Dhara Vikas program of the Rural Management and Development Department (RM&DD) of the Government of Sikkim. Steps 1 through 4 focus on knowledge generation and steps 5 and 6 on implementation. Table 3 summarizes the nature of data that are needed to identify the type of spring and manage it effectively. As it can be seen, some characteristics are measurable or quantifiable, while some other characteristics are qualitative.

The stressors are then identified by drawing a causal relationship between stressor variables and the state of the spring system. These stressors are characterized by measuring the change in the stressor variable while simultaneously quantifying the change in the 'state' of the spring. As classified earlier, we look at stressors as falling broadly into two categories, biophysical and socio-economical, though the effects are mostly combined, and sometimes the effect of one stressor in one category is on another stressor in the opposing category which finally impacts the 'state' variable of the spring. This interconnected nature of stressors should be kept in mind even while attempting to categorize the stressors. Characterization is conducted by identifying measurable properties (Table 4). For example, biophysical stressors are characterized by quantifying the change in climate variables such as precipitation and temperature at different temporal aggregations and scales, and by estimating the change in visible land cover year on year to quantify, for example, the decline in forest area, maximum areal extent of glacier fields and snowpacks, and canopy loss due to forest fires. Population increase, increase in the road and canal networks (reported as kilometers of roads constructed or canals excavated), increase in per capita demand of water to support upward economic and social mobility, etc., are some examples of properties of socio-economic stressors that can be quantified. Many of the stressors can be

characterized at a coarse spatial resolution from reports, data, and satellite imagery that are readily available. However, owing to the high spatially heterogeneous nature of the Himalayan landscapes, information at the local scale needs to be generated by (1) instrumentation within the springshed to measure, for example, rainfall and temperature, and by (2) adopting participatory methods at individual (key informant interview), household (household surveys), and community levels (village workshops, focused group discussions, transect surveys with community participants) to collect, for example, information on the use of natural resources (resource mapping, cropping calendars).

**Table 3.** Data requirements for identifying the type of springs.

| Sl. No. | Spring and Springshed Characteristic/Property | Data Required/Method Used | |
|---|---|---|---|
| | | **Measurable** | **Qualitative** |
| 1. | Geographic location | Latitude, longitude, elevation | |
| 2. | Administrative identifier | The village name, Gram Panchayat (lowest level of Panchayati Raj Institutions (PRIs) which form a part of the local self-governance system in India), district (an administrative division of an Indian state) | |
| 3. | Springshed boundary, area of interest | Toposheets, GIS, transect walk—all water sources inventoried with GPS | |
| 4. | Hydrology | Discharge, rainfall, flow duration curves, perennial or seasonal | |
| 5. | Hydrogeological mapping | Classification into spring type—depression, contact, fracture, fault, or karst types; identification of bedrock (regional aquifers) | Geological map of the area, observations during transect walk, conceptual hydrogeological layout (cross-section and 3D) of the springshed demarcating recharge areas |
| 6. | Water chemistry | pH, hardness, TDS, major constituents—$HCO_3$, $SO_4$, Na, Cl, Ca, Mg, Si; minor constituents—B, Ni, Co, K, F, Fe, strontium; trace constituents—heavy metals, nutrient concentrations (nitrates, phosphates ions), oxygen levels (DO, BOD) | |
| 7. | Human use—potable or not | | FGD, KII |
| 8. | Water uses | | FGD, KII, questionnaire survey—household survey; no. of households (from government records); no. of livestock (from livestock census); commercial uses; irrigated area |
| 9. | Importance of the spring | | KII—alternative sources of water, religious/cultural significance, perception of drying trends of the spring |
| 10. | Land use in the springshed | LULC maps—historical (baseline) and current | Transect walk, KII—cropping patterns, seasons, change in LULC |
| 11. | Management of the spring | | KII—conflicts and conflict resolution, existing institutions, equity aspects: women, marginalized sections, power dynamics |
| 12. | Access to the spring | | KII, questionnaire—private, few households, common, women only, marginalized sections |

Abbreviations: KII—key informant interview; FDG—focused group discussion; LULC—land use land cover.

**Table 4.** Measurable properties for characterization of stressors.

| | Drivers | Sources | How Is This Measured?(or) What Is the Variable That Needs to Be Measured? |
|---|---|---|---|
| **A** | **Biophysical drivers** | | |
| A.1 | Internal | | |
| A.1.1 | Deforestation—forested land converted to barren land | [6] | LULC map for at least two years (baseline/historical and current) and calculating change in the area of each LULC class. |
| A.1.2 | Replacement of multi-storied oaks with single-storied pine forests | [4,38] | Change in springshed aggregated evaporation losses from each LULC class. |
| A.1.3 | Change in the landscape in the recharge area—encroachment by invasive species (cacti and other xerophytes, Lantana camara, etc.) | [4] | Indicator for depletion in soil moisture. Percentage of area covered by oak forests replaced by pines or cacti and other xerophytes. |
| A.1.4 | Slope failures and landslides | [6] | Spatial distribution mapand density (landslides/$km^2$) of landslides developed after delineating existing landslides during fieldwork. |

**Table 4.** *Cont.*

| | Drivers | Sources | How Is This Measured?(or) What Is the Variable That Needs to Be Measured? |
|---|---|---|---|
| A.1.5 | Declining precipitation (particularly recycled precipitation recharging precipitation-fed springs) | [4] | Can be determined by a multivariate regression model with evapotranspiration and land use (categorical variable) as dependent variables and precipitation as independent variables. |
| A.1.6 | Forest fires | [15] | Thematic map of incidents of forest fires in the period under consideration (similar to landslide thematic map). |
| A.1.7 | The decreasing trend of precipitation in the period of snow accumulation, i.e., November to February, resulting in rapid glacier retreat due to negative mass balance, affects glacier-fed springs | [23,39] | Trend (or variation) in total precipitation received during November to February. |
| A.1.8 | Geomorphological features (aspect and slope) of the recharge area (upslope area) of the spring | [42] | Determined using Brunton compass during field survey. It can also be determined reliably from high-resolution DEM. |
| A.2 | External | | |
| A.2.1 | Precipitation decline due to climate change | [6] | The trend in annual precipitation. |
| A.2.2 | Reduction in the temporal spread of rainfall due to climate change, particularly decrease in winter rainfall | [15,21] | Trends in seasonal totals of rainfall. |
| A.2.3 | Increase in intensity of rainfall | [15,21] | Trends in the number of rainy days in a year. |
| A.2.4 | Shifting of climate zones to higher altitudes resulting in change reduction in oak forests and consequent replacement by pine and mixed forests | [6] | An indicator for shifting climate zones is % area covered by oak forests replaced by pines and mixed forest. Alternatively, change in mean elevation of oak forests, pine forests, and mixed forests. If the change is significant, then we can say that the climate zones have shifted. |
| A.2.5 | Glacier retreat due to global warming | [23] | Glacier retreat due to climate change is established using long records of glacier length (average record length used is 94 years) [43]. |
| A.2.6 | Neo-tectonic movements, active movements of tectonic plates resulting in changing thrusts, faults, and lineaments that affect groundwater flow | [6] | The proximity of spring location to seismo-tectonically active thrust planes, for example, the main boundary thrust. |
| **B** | **Socio-economic drivers** | | |
| B.1 | Internal | | |
| B.1.1 | Livelihood practices—uncontrolled grazing, frequent plowing for multiple cropping, and higher constructional activity in the recharge area affect the water holding capacity of the soil | [25] | The number of livestock per 1000 households (normalized by the number of households and area of the district) can be used as an indicator for pressure on land due to grazing. District-wise data of the livestock census are available for the 19th livestock census (2012).An increase in the built-up area, increase in abandoned agricultural plots, and increase in % of agricultural plots can be seen as indicators for livelihood practices that affect the sustainability of springs. |
| B.1.2 | Increasing fragmentation of landholdings. Therefore, springshed management requires greater coordination among more individuals who hold land in the recharge area. | [15] | Thematic map indicating the number of landholders per square kilometer similar to landslide map can be developed from household surveys. |
| B.1.3 | Development in the vicinity of the spring | [22] | The proximity of each spring to the nearest census town. The greater the proximity, the lesser the impact of the town on the spring. |
| B.1.4 | Road widening, excavation for roads | [4,6,22] | Key informant interview corroborated with village-level records. |
| B.1.5 | Expansion of canal network | [4] | Key informant interview corroborated with village-level records. |
| B.1.6 | Upslope cutting for settlement and big buildings | [6,22] | Village workshops and transect surveys to identify big construction projects and quantify area extent of upslope cutting. |
| B.1.7 | Encroachment into forest land for horticulture and agriculture | [4,6] | % pixel area of forest land replaced by cultivated land from LULC change analysis. |
| B.1.8 | Industrialization, urbanization | [4] | Increase in the built-up area from LULC change analysis |
| B.1.9 | Increasing population density in urban settlements, unplanned expansion, improper disposal of domestic sewage results in contamination of springs | [24,25,40] | Thematic maps of percentage increase in population between last two censuses. |
| B.1.10 | Out-migration and urban migration of men resulting in the feminization of springs management that is challenging local institutions | [41] | Household surveys and focused group discussions needed. |
| B.2 | External | | |
| B.2.1 | Development of densely populated settlements and large mountain towns in the same catchment as the springs under consideration, resulting in increasing demand for water | [5] | The proximity of each spring to the nearest census town. The greater the proximity, the lesser the impact of the town on the spring. |
| B.2.2 | Adoption of tourism models that are appropriate for plains in the IHR | [41] | Inventorying of villages with tourism as an alternative or primary source of livelihood. |

### 3.4. Impact of 'Stressors' on the 'State' of Indian Himalayan Springs

For the vulnerability assessment of Indian Himalayan springs, this section discusses the selection of critical 'states' based on 'stressors' identified in different regions of the Himalayas. The impacts can follow more than one pathway and are linked to a change in state. Table 5 summarizes the findings from the literature reviewed.

**Table 5.** Impact of 'stressors' on the 'state' of Indian Himalayan springs.

| Region | Stressor | Impact | State | Change in State |
|---|---|---|---|---|
| Sikkim, Central Himalayas | Climate Change<br>• Change in the temporal spread of rainfall<br>• Decrease in winter rainfall<br>• Increase in intensity | • Forest fires and resulting drying up of springs<br>• Poor accumulation and recharge during rainfall events resulting in the drying up of springs<br>• The decline in production of winter crops<br>• Women required to travel long distances to collect water | • Lean period discharge<br>• No. of months of flow | • Decrease in lean period discharge<br>• Change of perennial springs to seasonal or dried-up springs |
| Kashmir, Western Himalayas | • Winter precipitation—decreased rainfall in the period of snow accumulation<br>• Global warming | • Glacier retreat | • The maximum discharge of the spring in a year | • Attenuation of maximum discharge<br>• Karst springs are affected more than alluvial springs |
| Uttarakhand, Western Himalayas | • Climate change<br>→ The decline in precipitation from oceanic sources<br>→ → Shifting climate zones<br>→ Replacement of oak with pine<br>→ Conversion of forest land to barren land accelerated by clearing of forest for development and consequent erosion<br>→ Encroachment of alien species<br>→ The decline in recycled precipitation<br>• Economic and human development<br>→ Excavation for roads and canals<br>→ Construction activity<br>→ Reduction in the extent of forest cover due to overgrazing, encroachment into forest land for agriculture and horticulture, excavation for roads and canals, etc.<br>→ Unsustainable practices such as overgrazing and bi-monthly plowing resulting in loss of topsoil<br>→ Development of large towns and consequent pressure on water sources<br>→ Unmanaged drains and improper sewage disposal | • Decreased infiltration, increased surface runoff<br>• Decreased average seasonal discharge (in liters per day) in springs; the discharge measured periodically before the rainy season (June), immediately after the rainy season (October), and during winter (February)<br>• Diminished discharge in streams; annual average in cumec per day<br>• Springs in densely populated areas have high EC, low DO, high NO3-, and presence of coliform due to improper disposal of sewage | • Magnitude of discharge<br>• Seasonality<br>• Suitability of water for drinking | • Decrease in average discharge<br>• Decrease in the number of perennial springs and consequent increase in the number of seasonal and dried-up springs<br>• The presence of nitrates indicates contamination of the water |

### 3.5. Responses in Springs Due to Impacts and Stressors

Many spring revival programs have been successfully undertaken in the many Himalayan states during recent decades. The NITI Aayog report on Inventory and Revival of Springs in the Himalayas for Water Security [14] (pp. 23–24) lists appropriate responses to deterioration of springs based on a synthesis of best practices in spring revival in India. Recharge area protection or spring source protection has been adopted in many

locations by limiting grazing, encroachment, and development activities in the recharge area of the springs. An emphasis on the 'springshed' approach in policy and action has furthered the effectiveness of source protection by adopting hydro-geological methods to recharge area identification rather than a ridge-to-valley approach with just protection of the ridge or upslope area, as conducted in traditional watershed management. The effectiveness of the recharge of springs is improved by implementing engineering measures (staggered and contour trenching, building check dams across small streams, construction of gabions/retaining walls/spurs, diversion drains) and vegetative measures (fuelwood, fodder and fruit tree plantations, grassland management, and hedgerow treatment) [14]. Locations experiencing glacier retreats can adopt measures to improve snow retention and collection of snow meltwater for spring recharge. These measures have been seen to be successful when implemented through local-level management and adoption of traditional knowledge. Demand management is another response where communities dependent on springs calibrate their water consumption to ensure the sustainability of the springs. This involves improvement in water distribution, implementation of water conservation practices, and organized management of springs. A return to community ownership of the resource and strengthening of village-level institutions for water management has been observed in the success stories of spring revival. This ensures the implementation of interventions for spring revival that are linked to the livelihoods of the communities involved, thus further ensuring the long-term sustainability of the interventions. The dearth of long-term data on spring discharge and quality has hampered early identification and implementation of spring revival. There is a need for setting up systems for regular monitoring of springs, with at least one each from distinct typologies within an administrative unit (say, a district) for assessing the impacts and effectiveness of responses on the springs.

### 3.6. Framework Developed

The conceptual framework developed in this study aims to represent the vulnerability of springs in general (Figure 3) using novel systems thinking approaches. In the following figure, the target system comprises internal elements (Himalayan springs), and these elements are defined by characteristics (dotted circles within the system). Some examples of the spring characteristics are discharge, flow duration curve, and households dependent on a spring. Representation of springs can include multiple elements, including biophysical (BP), economic (E), and social (S) elements, or their combinations (BPES).

Stressors are external to the target system and are classified as internal and external stressors. They are defined by characteristics (dotted circles within the star). When the target system is exposed to the stressors, impacts are represented as changes in the characteristics of the target system elements (disfigured circles, color changes), representing changes to BP, E, S, and BPES. Note that each element responds differently (or not at all) to a stressor. On the other hand, the stressor characteristics are assumed to have no changes during impact. The small, medium, and large changes in spring characteristics are represented as changes in only color, only shape, and both color and shape, respectively. For example, a large increase in the number of heavy-rainfall days (stressor) may have a larger impact on discharge when compared to no change in discharge for a small increase in the number of rainy days in any year. Similarly, a large increase in the proportion of barren land in the recharge area may have a larger impact on the seasonality of the spring when compared to encroachment into forest lands for agriculture and consequent withdrawal of recharge water to the spring for consumptive water for farm produce. Accordingly, vulnerability assessments will be carried out to assess the degree of stress on the target systems. The efforts will be targeted to utilize the benefits of change and reduce the harmful effects of change. The thick arrows represent the overall flow direction of processes in the conceptual framework as well as the direction of the movement of time. Stressors and target system characteristics, the impacts of stressors, responses, and restora-

tion efforts can be identified by literature reviews, expertise in the region, and available observed/measured/survey data.

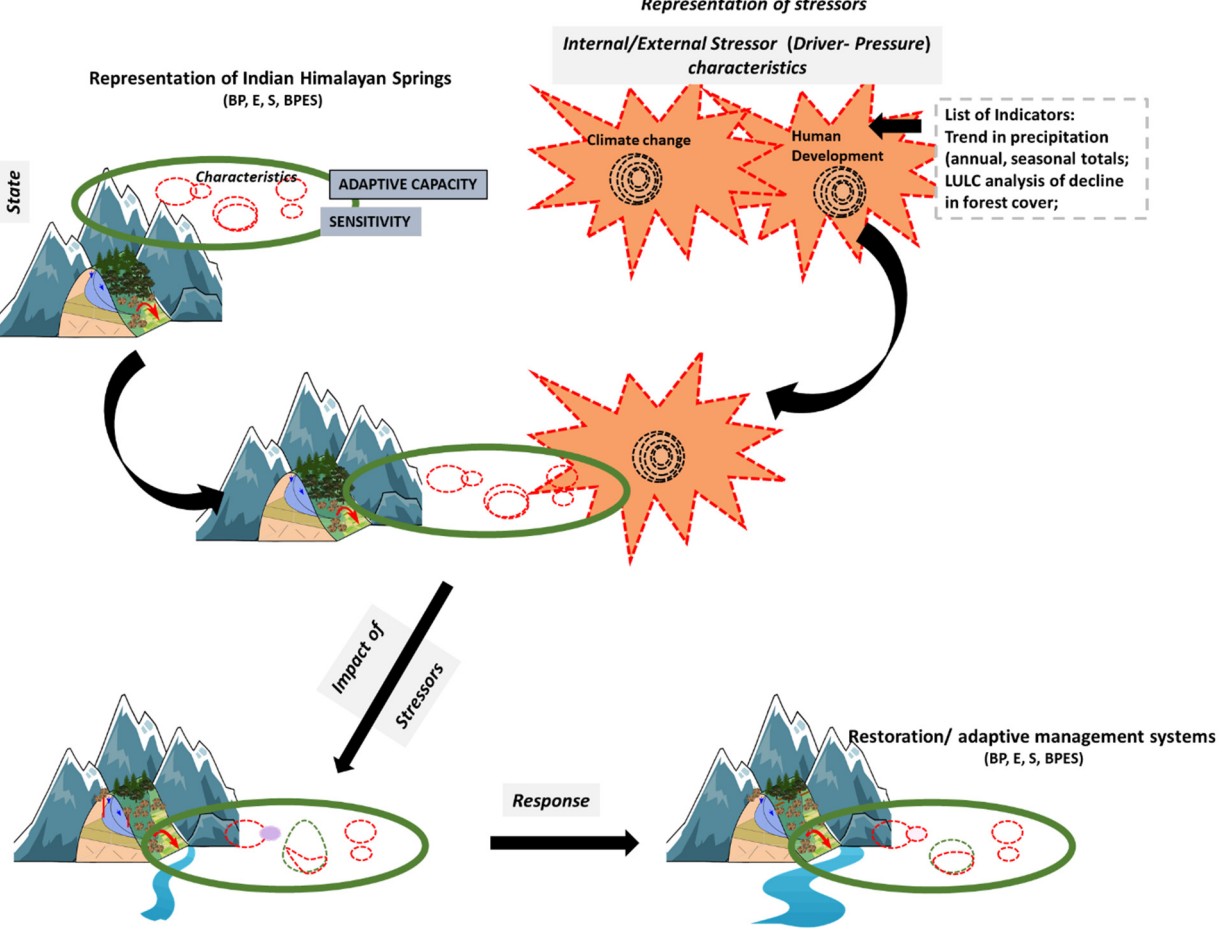

**Figure 3.** Conceptual framework.

## 4. Case Study

We present a limited stressor causal chain (Figure 4) using readily available data from previously published research for Mathamali spring in Dhanaulti Tehsil (administrative unit in India), Tehri-Garhwal district, Uttarakhand State of India. First, we look at the effect of biophysical stressors on springs. Analysis of annual and monsoon seasonal precipitation over a 60-year window (1955–2015) in the Tehri-Garhwal district shows a significant decreasing trend with the Theil-Sen slope reported as 8.22 mm/year and 6.80 mm/year, respectively [44]. It is important to note that this significant decreasing trend is higher than the annual and seasonal decreasing trend for the Uttarakhand meteorological sub-division (1.28 mm/year and 1.20 mm/year, respectively) based on an analysis of precipitation series for the years 1901–2019 [45]. A similar analysis of the most recent 30 years (1989–2018) of rainfall observations has also shown a decreasing trend, consistent with previous studies, in the annual and seasonal (southwest monsoon) totals of precipitation in Tehri-Garhwal district, but the reported trend is not statistically significant [46]. However, what was further reported in the 30-year analysis was a significant increasing trend in the number of heavy-rainfall days (daily rainfall >=6.5 mm) over the entire year, with no significant increase in the number of rainy days (daily rainfall >=2.5 mm) [46]. This implies that, on average in the most recent 30 years, over Tehri-Garhwal district and the springshed of Mathamali spring, there is an observed increase in the frequency of heavy-rainfall days, but the same (or slightly decreasing) annual and seasonal totals of precipitation deposited

over a water year. This is likely to result in less time for infiltration, increasing surface runoff, and, consequentially, less recharge for the same totals of precipitation. The increase in frequency in recent decades combined with the gradual decrease in precipitation over a great part of the previous century is a likely cause for the perceived decline in Mathamali spring's discharge.

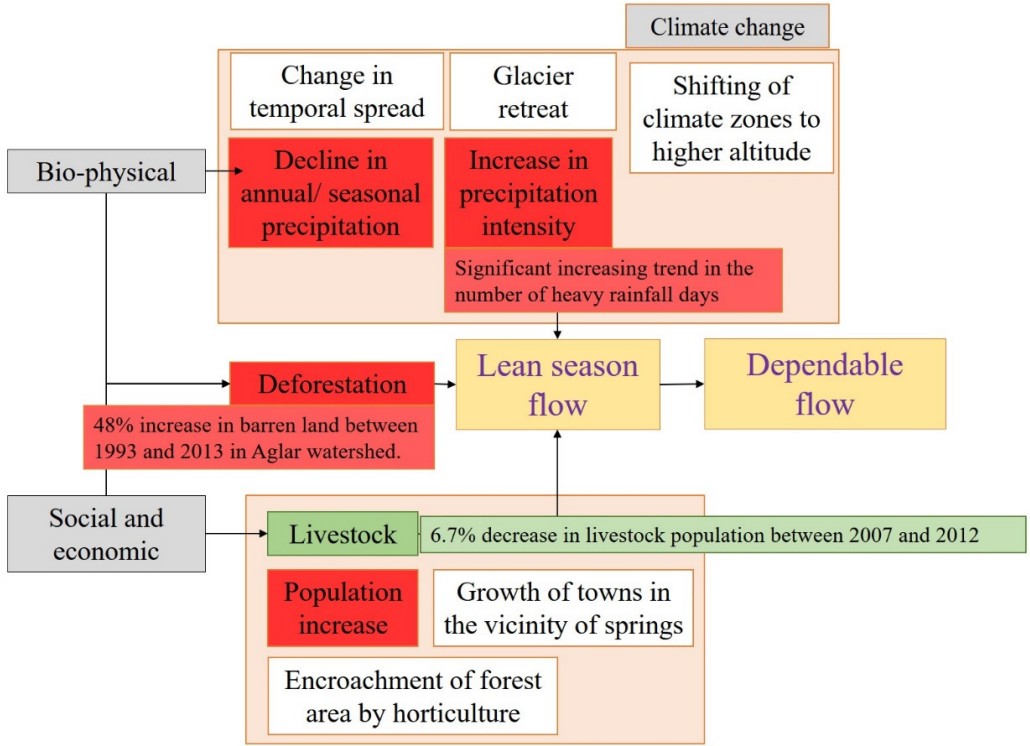

**Figure 4.** Characterization of stressors in Mathamali spring, Uttarakhand, India.

Furthermore, LULC analysis of the Aglar watershed [47], of which Mathamali spring forms a part, has shown a 48% increase in barren land between 1993 and 2017 and 19.8% between 1999 and 2017. Though the increase in barren land is very large in the early years of the time period, a conservative estimate of the decadal increase can be considered as closer to 10%. This is lower than the 28% increase in the fallow land in Uttarakhand State between 2007 and 2017, which increased from 108 thousand hectares in 2007–2008 to 150 thousand hectares in 2016–2017 (land use statistics, at a glance, for 2007–2008 to 2016–2017 published by the Ministry of Agriculture and Farmers Welfare, Government of India, November 2020, and accessed on https://eands.dacnet.nic.in/LUS_1999_2004.htm, accessed on 9 June 2021). An increase in barren land is likely to result in erosion and poorer retention of soil moisture, leading to a lower recharge of the spring.

Human and livestock populations exert pressure on the limited natural resources. We now look at the characteristics of human population and livestock population growth in Uttarakhand State vis à vis all-India growth numbers to identify and quantify the socio-economic stressors on the vulnerability of Mathamali spring. The average annual exponential growth rate of the human population in Uttarakhand for the decade 2001–2011 is 1.77, which is higher than the corresponding statistics of the IHR and pan-India, which are 1.71 and 1.64, respectively (2011 Census of India). The population growth rate being higher than other regions in the IHR indicates that the freshwater resources, including springs in Uttarakhand, are increasingly at risk of exploitation and consequent degradation (drying up of springs and contamination). Grazing practices exert pressure on the land, resulting in denudement of slopes in the recharge area. Between 2007 and 2012, the total livestock population in India decreased by 3.3% (18th and 19th livestock censuses of India). The corresponding decrease in the livestock population in Uttarakhand State is 6.7% (a decrease

from 51.41 lakh in 2007 to 47.95 lakh in 2012, as reported in 18th and 19th livestock censuses of India). This decrease in the livestock population in the 5-year period (2007–2012) must be seen in the context of a 5.6% decadal increase in India's total livestock population between 2003 and 2012 (17th and 19th livestock censuses). The recent decrease in the livestock population indicates that grazing pressure on land resources is not a significant stressor in the recent drying up of springs in Uttarakhand, yet the perceived drying up of Mathamali spring and the decadal increase in livestock indicate that the livestock population remains a socio-economic stressor on springs in general. The abandonment of agriculture (indicated by an increase in barren land) coupled with a growing livestock population and grazing practices (as a lucrative livelihood alternative to agriculture) puts further pressure on the denudement of slopes due to the combined effect of fallow ground in some locations and excessive grazing in other locations, thus affecting the recharge of springs and the decline in lean season flows. Therefore, in this case study of Mathamali spring, the biophysical stressors considered are the decline in annual and seasonal precipitation and the increase in barren land, while the socio-economic stressors are population and grazing pressures on land and water resources (Figure 4). The vulnerability index (*VI*) is estimated as a normalized aggregation of individual vulnerability indices (*VI$_i$*) due to each stressor, which is estimated using Equation (1). The equation is a modification of the similar equation suggested in [30] by trading time for space. Normalization is conducted by averaging the aggregated vulnerability index, aggregated using equal weights, but weights can be modified based on a detailed study of the impact of each stressor variable on the state of the spring. *VI* = 1 indicates a lower vulnerability of the spring to the stressors. The greater the *VI* deviates from 1 (*VI* < 1 and *VI* > 1), the greater the vulnerability of the spring of interest to the stressors compared to the springs located within the larger geographical extent. Here, by this definition of vulnerability, the effect of the stressors can be both detrimental (drying up of springs) and advantageous (increase in spring flow). The calculation of *VI* for Mathamali spring is demonstrated in Table 6.

$$VI_i = \frac{\text{Average value for the stressor variable in spring of interest}}{\text{Average value of the variable in the region (or country)}} \tag{1}$$

**Table 6.** Calculation of vulnerability index of Mathamali spring in Uttarakhand, India.

| Stressor Variable | Measured Quantity of Stressor Variable and Units | Average Value of the Stressor Variable in the Spring of Interest | Average Value of the Variable in the Region | Vulnerability Index |
|---|---|---|---|---|
| Precipitation | Decline in annual precipitation, mm/year | 8.22 | 1.28 | 6.4 |
| | Decline in seasonal precipitation, mm/year | 6.804 | 1.20 | 5.7 |
| LULC | Decadal increase in barren (fallow) area, percentage | 10 | 28 | 0.4 |
| Human population | Average annual exponential growth rate, % per year | 1.77 | 1.71 | 1.0 |
| Livestock population | Increase in livestock population between two censuses, percentage | −6.70 | −3.30 | −2.0 |
| | | Aggregated vulnerability index | | 11.5 |
| | | Normalized aggregated vulnerability index | | 2.3 |

The negative sign indicates that the effect of the stressor is in the opposite direction compared to other stressors. For example, a decrease in the livestock population moves the *VI* further to less than 1 (in the negative direction).

The vulnerability index of 2.3 in the limited case study indicates that the combined effect of the stressors is to decrease the water availability and deteriorate the water quality of freshwater from Mathamali spring in Uttarakhand. The effect of the decrease in precipitation is the dominant stressor (*VI$_i$* >> 1) affecting the decline in discharge of Mathamali spring, which, to a small degree, is mitigated by the decrease in the livestock population (*VI$_i$* < 0 and absolute value of *VI$_i$* > 1), both in the region and locally. The increase in barren land in the vicinity of Mathamali spring lowers the vulnerability of Mathamali spring (*VI$_i$* < 1) when compared to springs in the larger geographical extent of Uttarakhand State but still affects the decline in discharge, but to a lesser extent compared to other springs in the larger geographical extent. The estimation of a single quantity (*VI*) for vulnerability

is a useful metric for comparing springs and clusters of springs within a region to help prioritize spring rejuvenation efforts in a region.

The impact of the stressors on Mathamali spring is a possible decrease in lean season flow, reported as dependable flow. Discharge at Mathamali spring has been recorded at hourly time intervals and aggregated to daily averages from February 2014 to February 2018 [48]. Similar to other Himalayan springs, there was a general perception in the community that the spring discharge at Mathamali spring was declining. The spring was instrumented for continuous discharge monitoring to quantify the state of the spring and the impact of engineering measures and vegetative measures implemented to capture the limited precipitation in the soil matrix and improve the recharge to the spring. The 75% dependable flow showed a gradual increase from 5.7 in 2014 to 13.44 and 19.45 LPM in the water years of 2015 and 2016, respectively. The increase in dependable flows can be attributed to recharge interventions, both engineering measures, such as construction of a gabion check dam, multiple contour trenches, and a few artificial ponds (khals), and vegetative measures of tree, shrub, and grass planting, implemented since 2010–2011 by Garhwal Vikas Kendra in the recharge area of Mathamali spring (as reported in the Himmothan Society annual report 2010-11, accessed on http://www.himmotthan.org/, accessed on 9 June 2021).

This limited causal chain can be improved by including more stressors as information is made available, resulting in a more complex analysis of the vulnerability of Mathamali spring (Figure 5). The responses to changes in the 'state' of the spring are indicated in the causal chain by arrows leading away from the 'state' of the spring. For example, the response to declining flows is structural and vegetative measures in the springshed recharge area. The effectiveness of this response may be affected by the fragmentation of landholdings. Smaller landholdings will require the mobilization of a larger number of persons to adopt land management practices in their land parcels to help improve discharge in the springs. Declining water quality in springs is largely due to development activities in the vicinity of the spring, and the appropriate response will be better waste management and proper planning for future development in the region.

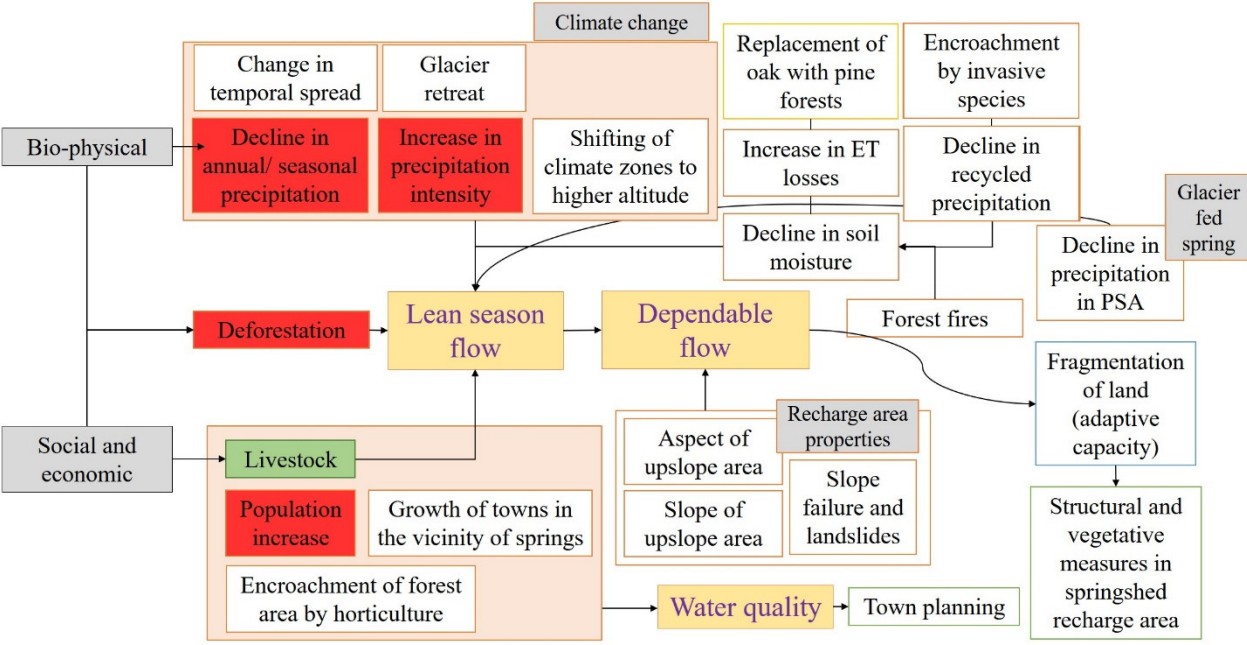

**Figure 5.** Vulnerability analysis of Mathamali spring, India.

## 5. Limitations in the Proposed Methodology

The main aim of this study was to propose a conceptual framework to represent the vulnerability of springs. The methodology enables the translation of theoretical constructs into actionable steps (operational procedure) and provides a take-off point for creativity in the use of indicators for quantifying vulnerability. Some immediate sources of error in the operational procedure for vulnerability analysis are presented in this section. First, the choice of stressors was conducted by deductive reasoning and constrained by the availability of easily accessible data. A suggested approach would be the use of regression models to establish the relationship between the state and stressors [49]. Second, constant weights were used in the normalization and aggregation of individual vulnerability indices. A correct choice of weights that reflects the correct interrelationship between the vulnerability stressors and the state of the spring is essential [30]. The weights can be improved through a consultative process of subject matter experts and stakeholders [27]. Thirdly, the present study demonstrated the use of readily available data (in a data-scarce region such as the HKH) for rapid assessment of vulnerability. In doing so, uniform time periods for analysis were not considered and could be a potential source of uncertainty. All datasets, as far as possible, should be suitably transformed to conduct the comparison of vulnerability stressors for the same time period (implying the same state of the system at the beginning of the time period). Fourthly, recent recharge interventions in the springshed comprising the spring may mitigate the effect of the stressors. This needs to be accounted for in the estimation of vulnerability to identify the springs which need further attention for springshed development and management. Finally, this study demonstrated an operational procedure for vulnerability analysis, the usefulness of which needs to be demonstrated in future work by using the procedure for prioritization of springshed development over a larger geographical region by enabling a ranking of all springs and clusters of springs within the region based on vulnerability to the stressors.

## 6. Conclusions

In this study, an attempt at conceptualizing vulnerability in Himalayan springs was carried out. First, the stressors (internal/external) affecting the springs were identified based on a systematic review of previously published research. Then, characteristics of the springs were identified, and causal links to the stressors were established by way of measurable quantities. By adopting a systems approach, a framework was then developed for representing the vulnerability of springs. This framework was demonstrated in the case study of Mathamali spring in the Western Himalayas. The case study was used to illustrate the operational procedure for vulnerability analysis. In the case study, vulnerability indices were computed for individual stressors, which were then aggregated and normalized to quantify the vulnerability of Mathamali spring. In the limited analysis, the dominant stressor affecting the vulnerability of Mathamali spring was identified as the decline in both the southwest monsoon (seasonal) and annual precipitation totals. The impacts and responses were synthesized from both the literature and expert knowledge.

The conceptual framework is a useful theoretical construct for enabling policymakers and project managers to follow a structured process for arriving at decisions concerning judicious use of resources towards a targeted springshed management of springs that are most vulnerable to biophysical and socio-economic stressors. This framework further addresses the pressing need for the synthesis of existing research findings related to Himalayan springs to extract information useful for decision-makers by capturing, visualizing, and organizing connections of a complex system through the conceptual framework. This study hopes to bridge the gap between the supply and demand of such useful information. With the impacts of stressors on springs becoming increasingly visible, identification of springs and clusters of springs requiring immediate attention has become necessary for efficient utilization of resources. The synthesis of stressors, identification of measurable indicators, and the conceptualization of vulnerability promote convergence towards

an evidence-based decision support system for rejuvenation and better management of Himalayan springs.

**Author Contributions:** Conceptualization A.A. and S.S.; methodology A.A. and D.D.; formal analysis D.D.; Writing—original draft D.D.; Writing—review and editing A.A. and S.S., Visualization A.A. and D.D.; Project administration S.S.; Funding acquisition A.A. and S.S. All authors have read and agreed to the published version of the manuscript.

**Funding:** This article is based upon work partially supported by National Science Foundation Grant no. 1735235 awarded as part of the National Science Foundation Research Traineeship, USDA-NIFA capacity building grant 2017–38821–26405, USDA-NIFA Evans-Allen Project, Grant 11979180/2016–01711, USDA-NIFA grant no. 2018–68002–27920 and the National Mission on Himalayan Studies (NMHS) Himalayan Research Fellowship Scheme grant number NMHS Grant: GBPNI.NMHS-2018-19/HSF31-09.

**Data Availability Statement:** No new data were created or analyzed in this study. Data sharing is not applicable to this article.

**Acknowledgments:** The authors wish to acknowledge Vikram Kumar who collected field data for Mathamali Spring during his Ph.D. in the Department of Hydrology, Indian Institute of Technology Roorkee, India during 2012–2018. Authors would also wish to acknowledge the contribution of David Behar, Director, Climate Program, San Francisco Public Utilities Commission, United States towards the conceptualization of the vulnerability model proposed in this work.

**Conflicts of Interest:** The authors declare no conflict of interest. The funders had no role in the design of the study; in the collection, analyses, or interpretation of data; in the writing of the manuscript, or in the decision to publish the results.

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
