# Peer review of "Conceptual Model for the Vulnerability Assessment of Springs in the Indian Himalayas"

_climate, doi:10.3390/cli9080121_

Round 1

Reviewer 1 Report

Authors present a conceptual model for the vulnerability assessment of springs in Indian Himalayan region. The paper reads well and can be accepted after minor revision. See the details:

  1. Authors affiliation didn’t follow the number correctly.
  2. More efforts require in Introduction section. One reference can’t cover the whole section. Search more literatures to back up.
  3. what are the pros and cons of top down, bottom up and hybrid approaches?
  4. conclusion section needs elaboration based on the main findings. 

Reviewer 2 Report

Dear authors:

The manuscript called "Conceptual Model for the Vulnerability Assessment of Springs in the Indian Himalayas" is a manuscript with medium scientific novelty. It is simple with respect to the methodology and hypothesis raised, which makes it easy to understand, and difficult to refute. Despite being a simple manuscript, I consider it a good scientific contribution due to the subject matter it covers (water problems in India, with a proposal to improve water savings). I recommend the manuscript for publication after minor revisions.

Start with a sentence that mentions the global water context:

The shortage of fresh water in various regions of the world is a major economic, environmental and social problem (

Cisternas, L.A.; Gálvez, E.D. The use of seawater in mining. Miner. Process. Extr. Metall. Rev. 2018, 39, 18–33, doi:10.1080/08827508.2017.1389729.

)

Must mention the shortage of spring water. This phrase can help:

Traditional sources such as lakes, groundwater and rivers, represent only 0.8% of the total water on the planet ( DOI: 10.1080/08827508.2017.1389729  ; DOI: 10.1016/j.jmrt.2019.11.020 )

Is there a record that indicates how water is consumed in the area? Is there an industry that generates overconsumption that affects the population?

orientation for 2500 “Km” along…. Correct to km

Regards

Reviewer 3 Report

The authors have developed a conceptual framework to evaluate the spring vulnerability in Indian Himalayas. They considered different stressors such as socio-economic, hydrology, biophysical, etc. in developed model. The conducted research seems interesting, particularly from water management viewpoint. I recommend the paper for publication after major and minor revisions described below are addressed.

Lines 30 to 35- The reference(s) is missing.  Please bring the relevant reference for stated numbers/paragraph.

Line 46 – Replace “is” by “are”

The introduction section requires significant improvements. The vulnerability of springs should be discussed with further details to highlight the importance of your study. Also, the variables affecting springs conditions in the selected region should be discussed, and clearly show how such variables affect water availability from springs and development of the stated framework. In summary, there is lack of reviewing sufficient literature in the introduction section.

Line 62- Replace “4” by “four”. Make the same change for “3” in line 63.

Line 96 – Add comma after “approach”

Line 156 & figure 2- How about considering land use change for socio-economic driver?

Line 184- Climate change was noted in the text and table 4 but not listed in Table 3 as required data for framework development.

Lines 308 to 317- The authors clearly analyze the selected case study. However, it would be interesting to also analyze the most important driver that affects spring condition. In other words, after reading the developed framework (figure 5) and its corresponding discussion, I did not get any final conclusion about the spring condition in the selected region. What would be the main factor controlling springs vulnerability in the selected case study?

Round 2

Reviewer 3 Report

Fortunately, the authors have addressed all my comments properly. My concern on lack of sufficient literature review has been addressed, and introduction section has been improved. The required missing references have been also added. 

Overall, I recommend the manuscript for publication.